# Analysis of Stigma in Relation to Behaviour and Attitudes towards Mental Health as Influenced by Social Desirability in Nursing Students

**DOI:** 10.3390/ijerph19063213

**Published:** 2022-03-09

**Authors:** Rosa Giralt Palou, Gemma Prat Vigué, Maria Romeu-Labayen, Glòria Tort-Nasarre

**Affiliations:** 1Faculty of Nursing and Physiotherapy, University of Lleida, 25003 Lleida, Spain; gloria.tort@udl.cat; 2SaMIS Group, Division of Mental Health, Althaia Foundation-UVic, 08243 Barcelona, Spain; gprat@althaia.cat; 3Adult Mental Health Center Horta Guinardo, Department of Public Health, Mental Health and Mother-Infant Nursing, University of Barcelona, 08007 Barcelona, Spain; mariaromeu@ub.edu; 4Health Education Research Group, Nursing and Phisioterapy Department, University of Lleida, 25003 Lleida, Spain

**Keywords:** attitudes, stigma, expected behaviour, social desirability, higher education, mental health, nursing students

## Abstract

The training undergraduate nursing students receive, both in terms of theoretical input and clinical practice, may help to instil a less stigmatising perception of mental health. To analyse the perceived evolution of attitudes and expected behaviours, a longitudinal repeated measures study was conducted in a population of student nurses during their undergraduate mental health education. The Mental Illness: Clinicians’ Attitudes Scale, a Scale for measuring attitudes to the mentally ill among future Health workers, and the Reported and Intended Behaviour Scale were completed. A mixed linear model was used to assess the effect of each factor in the questionnaires before and after the various stages of the students’ training in mental health. The overall effect of each factor was assessed by testing the interaction between factor and group, both with and without adjustment with the Social Desirability Scale. The results showed that the clinical practice stage, due to the proximity to care for people with mental health problems, improves attitudes and behaviours towards mental health in students who have not had mental health problems, and also in younger students. In conclusion, integrated, holistic training during the period of clinical practice was associated with positive changes in the attitudes and intended behaviour.

## 1. Introduction

Stigma is an attribute that has been analysed all through history. Etymologically, the word comes from Latin and Greek, where it meant a tattoo or a mark that might be applied to criminals as a punishment, or alternatively to social groups considered to have supernatural powers [1]. Over the years, the concept of stigma has gained prominence in many of the currents that study the impact of stigma on people with a mental health problem (MHP) [2]. Goffman (1963) defined it as a discrediting attribute, in which the external group discriminates against, devalues and punishes people affected by a mental health problem, and which obliges those affected to accept this discredit [3]. These attitudes and behaviours may be due to ignorance and prejudice [4], and according to the World Health Organization (2013), they affect all areas of the lives of people with MHP [5]. For this reason, the WHO argues that attention to mental health, defined as “state of well-being in which the individual realizes his or her own abilities, can cope with the normal stresses of life, can work productively and fruitfully, and is able to make a contribution to his or her community”, should be oriented towards mental health prevention, and should encourage social participation in the promotion and assessment of mental health in order to reduce stigmatization and discrimination. However, health professionals are often reluctant to take on this responsibility [6,7]. Within the health community, the nursing profession stands out for its direct and comprehensive approach to the care of mental health [8]. However, the lack of integrated knowledge regarding people with MHP and the skills required to treat them can in some cases generate attitudes and behaviours that favour stigmatization or overprotection, and fail to facilitate both the acceptance and the empowerment of persons with MHP and their participation in the health process itself [9,10,11,12]. This trend may be observed in undergraduate nursing students undergoing mental health training and may lead to ineffective learning, negative attitudes and behaviours, and social distance [13]. If this occurs, the nursing care given may not be sufficiently geared toward the recovery of people with MHP [14,15]. In contrast, in the best case scenario, nursing interventions can lead to a change of perspective with regard to stigmatizing attitudes and behaviours toward MHP [16]. Nursing professionals express more positive attitudes toward MHP as they gain experience [17,18]. Similarly, in university nursing students, training based on theoretical input can encourage favourable attitudes toward people with MHP. It also provides better preparation for later clinical practice [19], when interventions are provided directly with MHP. At this stage, due to the proximity of care, students’ anxiety and fears about mental illness are reduced [15,20,21], directing their attention towards acceptance and a positive attitude [19,22,23,24]. They also feel better prepared to perform specific mental health nursing procedures [25]. Thus, university training, based on the transmission of knowledge, the promotion of skills and attitudes integrated and focused on the person presenting MHP, can foster a positive change in the attitudes and behaviours regarding stigma among university nursing students [26,27]. 

However, it must be acknowledged that we live in social environments with their own norms, and that these norms frame socially desirable behaviours. This is how the tendency towards social desirability is forged [28,29]. According to Tourangeau and Yan (2007) social desirability is a dimension through which people’s responses reflect an attitude or behaviours that conforms to social norms [30]. In order to avoid social disapproval, people may avoid expressing their real opinions; thus, when analysing stigmatising attitudes and behaviours regarding mental health, social desirability may distort people’s responses and produce biased, unreliable data [31,32]. 

For all these reasons, in the present study, considering as a hypothesis that high quality university education will reduce stigmatising attitudes and behaviours towards care for people with mental health problems, the aim is to analyse the intended behaviours and attitudes towards MHP in a group of nursing students who receive training in mental health, taking into account the influence of social desirability.

## 2. Material and Methods

### 2.1. Study Design, Population and Setting

A longitudinal repeated-measures study carried out in a sample of 162 undergraduate nursing students from an original group of 180 students enrolled on a general mental health training program, studying at two universities in Catalonia, Spain.

The inclusion criteria were enrolment in the mental health nursing course, attendance at the induction session, provision of informed consent and the completion of the questionnaires. The students who only provided data at the first stage of the study were excluded.

Data were collected from September 2016 through June 2018.

### 2.2. Instruments

The questionnaire recorded socio-demographic data, including age, sex, history of a mental health problem, direct contact (present or past) with people with MHP, and previous professional experience and/or training in mental health. It also included scales measuring attitudes, intended behaviours and social desirability in relation to mental health.

#### 2.2.1. Mental Illness: Clinicians’ Attitudes Scale (MICA)

The MICA scale assesses attitudes toward mental health issues. Version 4 is specific to students and health professionals. It is scored on a Likert-type scale consisting of 16 items with six answer options, in which values of 1 indicate strong agreement and 6 strong disagreements. Overall scores range from a minimum of 16 to a maximum of 96, with higher scores indicating more stigmatizing and negative attitudes toward mental illness. The original scale had acceptable reliability (α ≥ 0.7). The validity indicated a moderate correlation with the items [33,34].

#### 2.2.2. The Spanish Scale for Measuring Attitudes to the Mentally Ill among Future Health Workers—Escala de Medición de Actitudes Hacia los Enfermos Mentales en Futuros Técnicos de Salud (Spanish Acronym EMAEMFTS)

This scale measures attitudes toward people with MHP among university students studying health degrees. It uses a 20-item Likert-type scale, with five response options ranging from 0, “strong agreement” to 4, “strong disagreement”. The overall score ranges from 0 to 80, with higher scores indicating more positive attitudes toward mental health. The scale’s reliability is α 0.87, and its validity is high [35]. 

#### 2.2.3. Reported and Intended Behaviour Scale (RIBS)

This scale assesses the presence of reported and intended behaviours in the general population in different mental health contexts, e.g., being with, working with, living close to, and having a relationship with a person with MHP. It is scored on a 5-point Likert scale, with five answer options ranging from 1 = strong disagreement and 5 = strong agreement. The overall score ranges from 0 to 20, with higher scores indicating more favourable expected behaviours. As regards its psychometric properties, it has a reliability of α 0.75, and high validity [36,37].

#### 2.2.4. Social Desirability Scale (SDS)

This scale explores the tendency in respondents to modify their answers in order to present themselves in a favourable light, based on convenience and conformity with social norms.

The scale consists of 33 items, with two answer options (true and false). The overall score ranges from 0 to 33, with higher scores indicating higher social desirability. As regards its psychometric properties the original scale had a reliability of 0.70–0.85 [28,38].

### 2.3. Ethical Considerations

The study was approved by the Ethics and Research Committees of the universities (file No 07/2016). Prior to the start of the study, potential participants were invited to an information session at which the study was described. Care was taken to ensure that the information given did not generate stigmatizing tendencies or any social bias.

Participants who agreed to enter the study gave their written consent immediately prior to the administration of the instruments. Confidentiality and anonymity were guaranteed.

### 2.4. Procedure

Data were collected at three different time points. In the first (L0), in September 2016, students had not yet started their theoretical training. In the second (L1), in December 2016, they had completed their theoretical training, and in the third (L2), from January 2017 to June 2018, they had completed their clinical placement period.



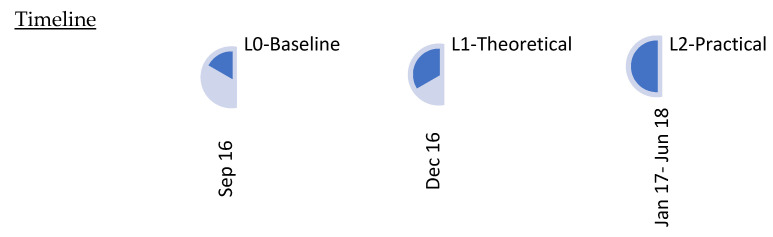



At each time point, the questionnaires were administered individually to each student, after informing them of the purpose of the study. Students were also told that participation was voluntary and anonymous, and would not influence their academic outcomes.

### 2.5. Statistical Analysis

In the descriptive analyses, means and standard deviations were used for continuous variables and frequencies and percentages for categorical variables. Cronbach’s alphas were calculated to measure the internal consistency of the questionnaires before and after each period of mental health training (baseline, post-theoretical input and post-clinical practice).

For each questionnaire, linear mixed models were used, performing one-way ANOVA for statistical assessment of the differences in groups. Pairwise comparisons were used to determine which group differences were statistically significant.

A linear mixed model was used to assess the effect of each factor on the questionnaires before and after each period of mental health training. The factor, group and factor-by-group interaction were included as fixed effects, and students were included as random effects of the different repeated measures. The overall effect of each factor was assessed by testing the interaction between factor and group, both with and without adjustment with the SDS.

Finally, factors and factor-by-group interaction found to be associated in each questionnaire were entered into multiple linear mixed models. The final models were performed adjusting for the SDS, estimating the regression coefficients and their 95% confidence intervals.

The restricted maximum likelihood calculation method was used in the models. The appropriate covariance structure for the data was determined using the Akaike Information Criterion. A two-sided type I error of 5% was considered statistically significant. Data were analysed using IBM SPSS Statistics for Windows version 25 (IBM Corp, Armonk, New York, NY, USA) and R version 3.3.2 (R Foundation for Statistical Computing, Vienna, Austria).

## 3. Results

### 3.1. Participants Characteristics

The main socio-demographic characteristics were recorded for the 162 students included (i.e., 90% of the study population initially recruited). Mean age was 23 years (± 5.9), and 75.3% were women.

Almost two-thirds of the sample (66.0%) knew someone close to them who had or had had a mental health problem, while only 8.6% had or had had a mental health problem themselves. In all, 11.1% of the sample had previous professional experience in mental health, and 10.5% had previous training experience in mental health (Table 1).

### 3.2. Evolution of Scores in Attitudes Based on Participants Characteristics and Mental Health Training

Table 2 presents the overall scores of the means, standard deviation, and differences between groups before and after each period of mental health training. Significant overall differences were shown in each of the scales studied, as well as when comparing each progressive period of university training, pointing to a progressive reduction in stigmatizing attitudes and an increase in favourable intended behaviour.

Table 3 shows the internal consistency of the scales.

The internal consistency values obtained on all the scales were similar to those of the original versions. Thus, it can be considered that the instruments used ensure reliable results.

### 3.3. Associations between Participants Characteristics

Each variable was associated with each of the factors both adjusting and not adjusting for the SDS. In this case, the associations were significant with respect to the RIBS scale in students aged 21 or less, and after the completion of the theoretical input stage (*p* 0.004). In addition, in the case of the EMAEMFTS scale, significant associations were reported during the period of clinical practice and in students who had not presented an MHP (*p* 0.003). In both scales the results are maintained regardless of adjustment of the SDS. (Table 4). Figure 1 shows the models with statistically significant interactions.

## 4. Discussion

This study examines the intended behaviour and attitudes toward mental health among undergraduate nursing students over the course of their training in the field of mental health. We believe that this is the first study in our country to assess these concepts among nursing students also taking account of the influence of social desirability.

The evolution of the students’ overall scores over the course of their training period presented significant improvements in both future intended behaviours and attitudes toward people with MHP. These results are in line with those obtained by Clement et al. (2012) when analysing different training strategies in nursing students, such as watching recordings of service users/informal carers relating their experiences or watching live presentations by people with MHP [39]. Attitudes also improve during the training process in medical students [40,41] and in health professionals from different disciplines who treat patients with mental health issues [42].

As for the factors associated with attitudes, the EMAEMFTS scale showed an association between positive attitudes and the absence of MHP after the completion of the clinical placement period, both with and without adjustment for social desirability bias measured by the SDS. The use of this instrument created and validated in the same country, although considered a strength of this research, the evidence sought points to the non-existence of other studies where this scale is considered, so the results cannot be contrasted. Studies using other scales have indicated acceptance and positive attitudes toward mental health after contact with people with MHP [43,44,45]. 

Other studies suggest a relationship between the training received and less stigmatising attitudes of student nurses, providing a more positive experience in relation to mental health care [46]. Better acquisition of knowledge, attitudes and future behaviour was also reported when university teaching plans for mental health include the issue of stigma [47]. However, one of the key components of the training in mental health given to nursing students is the integration of conceptual learning through clinical practices. In clinical placements, through the closeness of nursing care, interest in and empathy for individuals with MHPs is enhanced [23,48]. Unfortunately, not all clinical practice environments reinforce the knowledge already acquired: in one study, participants who carried out their clinical practices in a non-mental health setting did not demonstrate more favourable attitudes toward mental health nursing [20]. For this reason, practice environments must promote the acquisition of appropriate skills in order to improve behaviours and attitudes toward people with MHP [49,50]. 

The clinical placement period also had a significant effect on the expected behaviour of our sample of students aged 21 or under, and do not suffer and have not suffered from any MHP, regardless of social desirability. This trend can be interpreted as younger students also being more malleable, improving more easily in new learning, and extrapolating these into their own personal and professional attitudes [47]. The current study provides new information regarding the positive impact of direct contact with MHP on young nursing students. This trend is also seen in the general population [51], but not in all samples of nursing students [47] or professional nurses [52,53]. 

One of the strengths of this research is based on the fact that one of the characteristics analysed was the evolutionary process of stigmatising attitudes and behaviours towards MHPs, due to the fact that they suffer or have suffered from a MHP, taking into account the influence of social desirability. According to the literature reviewed, it has not been previously investigated in university nursing students [18,23,45,47]. It has also not been studied in the general population [37,51].

In our study we did not find significant results when analysing the variables on the MICA scale. Previous reports using this scale have indicated lower stigmatizing attitudes both in nursing students whose training includes direct social contact with people with MHP [39] and in health professionals receiving new theoretical input in mental health [53]. In addition, in psychiatry students, the results suggested a decrease in stigmatizing attitudes when participants had MHP or had friends or family in this situation [41]. Those results were consistent with those reported by Hernandez Arroyo et al. (2015), in a population of medical students who had not yet had direct contact with people with MHP, and by Dalky et al. (2020), with health professionals with less knowledge of mental health and less direct contact with people with MHP; in those studies, the authors recorded an undesirably high level of stigmatizing attitudes [34,52]. 

Gender was also not a significant variable in our research. However, previous reports have indicated gender-related differences with regard to stigma, finding female gender to be a protective and destigmatizing factor both in the nursing environment [53] and in the general population [54]. However, Aznar-Lou et al. (2016) found more positive attitudes among males [51]; Dalky et al. (2019) suggested that gender may generate social desirability bias [52]—as did Rasinski et al. (1994), who found that when asked questions by interviewers and in the presence of relatives women were less likely to tell the truth [55]. 

These results suggest the need to continue emphasizing the value of direct contact with people with MHP in undergraduate nursing training in mental health. Petkari et al. (2018) also identified the battle against stigma and social distance as key elements in students’ clinical practice periods [56]. One of the new tools for encouraging the integration of knowledge is the participation of expert patients (Experts by Experience) in mental health training plans [57]. Introducing this change into university frameworks opens up a new perspective, and may enhance nursing care and interventions focused on the recovery and autonomy of people with MHP.

### Limitations of the Study

As this study included only students at Catalan universities, the findings may not be generalizable to other contexts. Not all sociodemographic characteristics of the participants were considered: possibly, influential response factors may have been overlooked, such as socioeconomic status, ethnic characteristics, level of prior knowledge, level of prior professional experience, and degree of closeness to mental health.

At the time of data collection, although participants had been told that their participation would not affect their academic performance, the fact that they were enrolled in a mental health training course may have made them feel compelled to participate. However, data collection was voluntary, anonymous, and coded.

Finally, there may have been a self-selection bias, with only the most motivated students participating in the study. Those who declined to participate were not required to give their reasons.

## 5. Conclusions

In undergraduate nursing students, training in practical settings in which the focus is on personal care may achieve positive changes in attitudes toward mental health and foster non-stigmatizing values and behaviours.

Our results may be relevant to the design of nursing curricula, consolidating the integration of theoretical input and skills with practice-based learning. The recognition of the possible effects of social desirability bias also increases the authenticity of the responses regarding attitudes and intended future behaviour.

### Relevance to Clinical Practice

The main contribution of this study is its authentic and sincere reflection of the attitudes of nursing students to mental health care. It also identifies the stages in the training period that are most conducive to achieving positive changes in attitudes and future intended behaviours in relation to mental health. The results may provide guidance on the training itineraries and input that should be intensified in order to reduce stigmatizing attitudes toward people with MHP.

## Figures and Tables

**Figure 1 ijerph-19-03213-f001:**
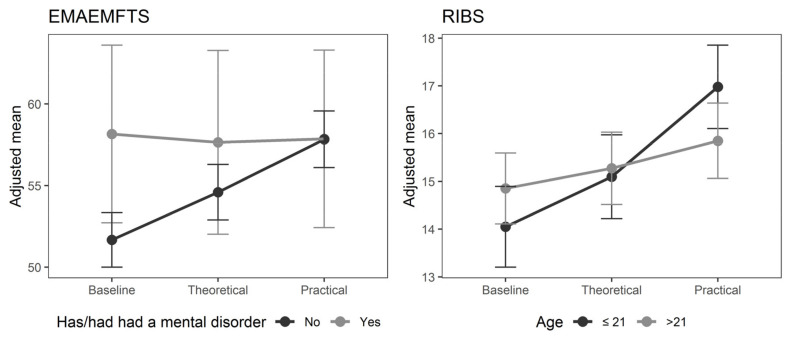
Models with statistically significant interactions. Spanish acronym EMAEMFTS: Scale for measuring attitudes to the mentally ill among future health workers; RIBS: Reported and intended Behaviour Scale. Baseline = L0; Theoretical = L1; Practical = L2.

**Table 1 ijerph-19-03213-t001:** Participants characteristics.

Characteristic	*n* = 162
**Age**	23.0 (SD 5.9)
**Gender**	
Female	122 (75.3)
Male	40 (24.7)
**Contact with people who have or had a mental disorder**	
No	55 (34.0)
Yes	107 (66.0)
**Has or had had a mental disorder**	
No	148 (91.4)
Yes	14 (8.6)
**Previous professional experience**	
No	144 (88.9)
Yes	18 (11.1)
**Previous training experience**	
No	145 (89.5)
Yes	17 (10.5)

Data presented as: mean (SD); *n* (%).

**Table 2 ijerph-19-03213-t002:** Global scales results and differences between groups before and after each period of mental health learning.

	L0	L1	L2	*p*		*p*	
	Mean (SD)	Mean (SD)	Mean (SD)	General	L0 vs. L1	L0 vs. L2	L1 vs. L2
**MICA**	40.6 (7.7)	39.9 (9.2)	37.7 (8.1)	0.001	0.633	0.003	0.010
**EMAEMFTS**	52.3 (10.4)	55.0 (10.4)	58.0 (10.1)	<0.0001	0.002	<0.0001	0.0002
**RIBS**	14.7 (3.6)	15.4 (3.7)	16.5 (3.3)	<0.0001	0.071	<0.0001	0.0002

Data presented as: mean (SD), *p* ≤ 0.05. MICA: Mental Illness: Clinicians’ Attitudes Scale; Spanish acronym EMAEMFTS: Scale for measuring attitudes to the mentally ill among future health workers; RIBS: Reported and intended Behaviour Scale; SDS: Social Desirability Scale. Baseline = L0; Theoretical = L1; Practical = L2.

**Table 3 ijerph-19-03213-t003:** Cronbach’s alpha.

		α	
	L0	L1	L2
**MICA**	0.617	0.71	0.644
**EMAEMFTS**	0.859	0.876	0.873
**RIBS**	0.700	0.762	0.759
**SDS**	0.773	0.799	0.808

MICA: Mental Illness: Clinicians’ Attitudes Scale; Spanish acronym EMAEMFTS: Scale for measuring attitudes to the mentally ill among future health workers; RIBS: Reported and intended Behaviour Scale; SDS: Social Desirability Scale. Baseline = L0; Theoretical = L1; Practical = L2.

**Table 4 ijerph-19-03213-t004:** Mean for participants characteristics and related to social desirability-adjusted mental health exposure.

									Unajusted			Adjusted by SDS	
		L0		L1		L2			*p*			*p*	
		Mean	SD	Mean	SD	Mean	SD	Time	Categ	Time x	Time	Categ	Time x
**MICA**										**Categ**			**Categ**
Gender	Male	42.2	8	43.5	10.7	42.4	7.9						
	Female	41	8.1	39.1	9.1	37.3	7.5	0.000	0.110	0.255	0.000	0.050	0.239
Age	≤21	39.5	7.1	38.9	8.4	36.4	8						
	>21	41.4	8.1	40.1	9.8	38.8	8	0.000	0.077	0.889	0.000	0.054	0.878
Contact with people who have or	No	40.4	7.9	40	10.9	37.7	8.3						
had a mental disorder	Yes	40.6	7.6	39.8	8.4	37.7	7.9	0.000	0.983	0.916	0.000	0.619	0.911
Has or had had a mental disorder	No	40.5	7.5	40.3	9.3	38	8.2						
	Yes	40.7	9.6	35.2	8.1	35.3	5.5	0.000	0.259	0.198	0.000	0.144	0.282
Previous professional experience	No	40.7	7.3	40	9	37.8	8						
	Yes	39	10.2	38.5	11	36.7	8.6	0.000	0.419	0.829	0.000	0.469	0.828
Previous training experience	No	40.7	7.6	39.8	9.2	37.5	8						
	Yes	39.1	8.2	40.1	10.1	39.6	8.2	0.000	0.984	0.252	0.000	0.936	0.260
**EMAEMFTS**													
Gender	Male	50.7	9.9	51.2	10.8	52.8	11						
	Female	53.4	11.4	56.3	10.7	58.5	10.1	<0.0001	0.124	0.169	<0.0001	0.071	0.156
Age	≤21	52	9.7	55.5	9.9	59.4	9.3						
	>21	52.3	11	54.5	10.9	56.8	10.6	<0.0001	0.659	0.172	<0.0001	0.590	0.166
Contact with people who have or	No	50.5	10	54	11	57.7	11						
had a mental disorder	Yes	53.2	10.5	55.5	10.3	58.1	9.7	<0.0001	0.275	0.309	<0.0001	0.140	0.353
Has or had had a mental disorder	No	51.8	9.8	54.6	10.3	58	10.1						
	Yes	57.5	15	59.6	12	57.2	10.3	<0.0001	0.299	0.033	<0.0001	0.201	0.033
Previous professional experience	No	51.9	10.4	55.0	10.5	57.8	10.3						
	Yes	55.7	10.5	55	10	59.2	8.5	<0.0001	0.366	0.360	<0.0001	0.398	0.316
Previous training experience	No	52.2	10.6	55.2	10.6	58.2	10.4						
	Yes	53.3	8.6	52.8	9	56	6.7	<0.0001	0.805	0.403	<0.0001	0.767	0.405
**RIBS**													
Gender	Male	14.9	3.9	14.2	4.6	5.2	4.2						
	Female	15.1	3.4	16.1	3.2	16.3	3.4	<0.0001	0.211	0.077	<0.0001	0.122	0.064
Age	≤21	14.2	3.8	15.3	3.6	17.1	2.8						
	>21	15.1	3.5	15.4	3.8	16	3.6	<0.0001	0.980	0.004	<0.0001	0.937	0.004
Contact with people who have or	No	14	3.5	14.7	4	16.2	3.6						
had a mental disorder	Yes	15.1	3.7	15.7	3.6	16.7	3.2	<0.0001	0.086	0.620	<0.0001	0.028	0.671
Has or had had a mental disorder	No	14.5	3.6	15.2	3.7	16.5	3.3						
	Yes	17	3.2	17.2	2.6	17	3.9	<0.0001	0.078	0.117	<0.0001	0.039	0.104
Previous professional experience	No	14.6	3.5	15.5	3.6	16.6	3.2						
	Yes	15.5	4.5	14.4	4.2	15.8	4.5	<0.0001	0.797	0.089	<0.0001	0.729	0.073
Previous training experience	No	14.7	3.5	15.6	3.5	16.6	3.2						
	Yes	14.35	5.1	13.7	5	15.4	4.3	<0.0001	0.159	0.356	<0.0001	0.140	0.351

Data presented as: Mean (SD); *p* ≤ 0.05. MICA: Mental Illness: Clinicians’ Attitudes Scale; Spanish acronym EMAEMFTS: Scale for measuring attitudes to the mentally ill among future health workers; RIBS: Reported and intended Behaviour Scale. Baseline = L0; Theoretical = L1; Practical = L2.

## Data Availability

Data is contained within the article.

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
