# Peer review of "Analysis of Stigma in Relation to Behaviour and Attitudes towards Mental Health as Influenced by Social Desirability in Nursing Students"

_ijerph, 2022, doi:10.3390/ijerph19063213_

Round 1
Reviewer 1 Report
Thank you for submitting a manuscript, entitled “Intended behavior and attitudes toward mental health in undergraduate nursing students, and the influence of social desirability”. Here comes with my comments below.
Abstract
- The background of the study should be briefly explained. It’s inadequate to describe it and the aims of the study should be illustrated. AT the end, although clinical practice may be the factor affecting the attitudes and behavior, what reason in the clinical practice made the results.
- Past tense should be used .
Introduction
- Any reasons the last paragraph was bolded.
- The introduction was inadequately described. Knowledge gap should be clear to let the readers understand why your study was important.
2.3 Procedure
- The Authors should elaborate more about the procedure. I don’t understand why the clinical practice was involved and was the important reason to explain students’ attitudes and behaviors.
Results
- Tables should be followed the respective parargraph.
- Table 2, the p values were unclear. What analytic methods were conducted. If necessary, separate the results if different analytic methods were used for different purposes.
- Same as for Table 4. It’s difficult to read the results of the end columns
Discussion
- Authors should focus on the result interpretation such as the significant results on MICA, EMAEMFTS and RIBS.
- Why were those characteristics, such as age, gender, contact with people who have or had a mental disorder…., being analyzed? The authors should elaborate more the results in the table 4
References
Need to follow the journal instruction on referencing.
Overall
- English proficiency should be changed.
- The journal format should be followed.
Author Response
Thank you for your interest in participating in the review of this article. The contributions made have been very useful to improve the work and final result.
The authors of the article wish to return the suggestions provided, and remain at your disposal for further suggestions if necessary.
As you suggested, the summary has been remodeled in order to improve its global comprehension. We have also corrected the verb tense, which in some paragraphs did not match the required one.
In the Introduction section, the hypothesis of the work has been included in addition to the objective of the study.
The reason why the first paragraph of the text was in black is unknown. It has been verified that this was not the case in the original document that was sent.
As for the Procedure, the text has been revised, although keeping the same subsections in which the work was already arranged in the original presentation. It is considered that in this way it correctly describes each of the relevant characteristics of the research, as well as the context in which it was developed.
Regarding the comment related to the clinical practice and its implication in the response of attitudes and behaviors, it is considered that the clinical practice is one more stage in the collection of research data, as is reflected in the subsection 2.4 and in the attached timeline.
In the Results section, the location of the tables has been restructured as required by the journal itself.
In each table, no new information has been provided with respect to the methods used in the data analysis, as it is considered repetitive, being reflected in the information corresponding to section 2.5.
Regarding table 4, it has been newly attached, improving the visibility and readability of the same.
As for the section referring to the Discussion, the contents have been revised and readjusted, referring to the motives and the analysis of the discussion corresponding to the data obtained, introducing new research that refer to and contrast the same.
The references have also been adapted to the publication characteristics required by the IJERPH journal.
Best regards
Reviewer 2 Report
Comments to the Authors are in the attachment.

Author Response
Thank you for your interest in participating in the review of this article. The contributions made have been very useful to improve the work and final result.
The authors of the article wish to return the suggestions provided, and remain at your disposal for further suggestions if necessary.
We have taken into account the improvement regarding the title of the research, including the term stigma, since, as you indicated, it better describes the analysis aspects of the research itself.
Regarding the Introduction, the text has been revised, and the terminology has been unified, orienting it in a clearer way towards people who present a mental health problem. It has also described more clearly what is meant by Mental Health. The objective of the study has been adjusted, also providing the hypothesis of the study. We believe that this has improved its understanding.
As for the Methods section, the presentation of the indicated subsections has been reordered.
As for the reading of the Results, the presentation has been structured, as indicated in the presentation guidelines of the journal, including each table together with the corresponding text. Part of the content of the text has also been partially retouched, although the authors have considered not to extend it so as not to repeatedly introduce information already explained in section 2.5.
As suggested, the title of the presentation of table 1 has been retouched, avoiding confusion regarding the data provided in the same. The visualization of table 4 has also been improved.
Regarding the other comments provided, the type of letter has been unified, which by mistake was not detected at the time of sending the work to the journal. The bibliography has also been adapted to the editorial criteria of the journal.
Best regards
Reviewer 3 Report
General comments:
Thank you for opportunity for reviewing this interesting paper. Research partially adheres to CONSORT guidelines. I believe that the topic of the manuscript is very interesting.
I believe that this manuscript doesn´t qualify for acceptance at this time and should be improved for publication in IJERPH.
Specific comments:
- Writing
The writing, structure and organization of the manuscript is in accordance with the guidelines.
- Title
The title reflects the content and problem studied.
- Abstract
The abstract reflects reflects the manuscript and provide an informative and balanced summary of what was done and what was found
- Key Words
The keywords are representative of the subject studied and exposed. Mesh Terms is social stigma
- Background
The background reflects the state of the art in relation to the study. The objective of the study is mentioned, as well as the justification for the choice and importance of studying this theme.
The last paragraph is in bold type
- Methods
There is detailed description of the research methods used. The design is correct and it is possible to validate the veracity of the results
They describe the setting, locations, and relevant dates, including periods of recruitment, exposure, follow-up, and data collection.
Eligibility criteria and instrument were explained in detail.
- Findings
Authors should consider using a flow chart.
In the results I would like to see the description of the most important results with the table support
Tables
Table 1. Age. 23.0(sd 5.9). No need to put SD
Table 4 is not visible
- Discussion
The key results of the discussion are concrete. In addition, it includes the main strengths and weaknesses in relation to other studies, discussing important differences in the results.
Limitations haven´t been exposed
It´s clear and concise. the conclusions are in line with the objective.
- References
The references used are correct, the vast majority dating back less than ten years.
Author Response
Thank you for your interest in participating in the review of this article. The contributions made have been very useful to improve the work and final result.
The authors of the article wish to return the suggestions provided, and remain at your disposal for further suggestions if necessary.
As suggested, the publication guidelines on which the IJERPH journal is based have been reviewed, modifying and adapting the contents, considering that at this moment they are correctly completed.
The reason why the last paragraph was underlined in black is unknown. It has been verified that this was not the case in the original document that was sent.
The tables have been attached to the text, making it easier to read and understand.
In relation to the suggestion of withdrawing the SD values, the standard deviation of table 1 has not been eliminated, since it is considered that it could provide information that could be interesting for the description of the selected population.
The limitations of the study are included in a specific section after the discussion.
Best regards
Round 2
Reviewer 1 Report
Thank you for submitting the revised manuscript. This version has been written more clearly.
2.1 "A' longitudinal....
good to proceed to publish.
Author Response
Thank you for your quick review, as well as for your feedback on the changes made to the article.
The new comments have been taken into account. Minor spelling errors have also been checked and corrected.
The authors are again at your disposal for further suggestions if necessary.
Sincerely
Reviewer 3 Report
Thank you for the opportunity to revisit this interesting article. The research adheres to CONSORT guidelines.
I think the topic of the manuscript is very interesting and now meets the requirements to be accepted at this time and be enhanced published in IJERPH
Author Response
Thank you for your quick response in reviewing the article. The authors thank you for your feedback and your contribution to the improvement of the article.
Some spelling mistakes in the text have been corrected.
Yours sincerely